# Beyond Text-Only: Towards Multimodal Table Retrieval in Open-World

**Da Li**[1,2,3]    **Keping Bi**[1,2,3 †]    **Jiafeng Guo**[1,2,3 †]    **Wei Yuan**[4]
**Fang Yang**[4]    **Tingting Gao**[4]    **Xueqi Cheng**[1,2,3]
[1] State Key Laboratory of AI Safety
[2] Institute of Computing Technology, Chinese Academy of Sciences
[3] University of Chinese Academy of Sciences   [4] Kuaishou Technology
{lida21s, bikeping, guojiafeng, cxq}@ict.ac.cn
{yuanwei05, yangfan}@kuaishou.com

## Abstract

Open-domain table retrieval aims to retrieve semantically relevant structured tables from a large-scale corpus in response to natural language queries. Unlike unstructured text, tables store information not only through their textual or numerical content but also through their structural properties, including hierarchical relationships between headers and cells, as well as complex spatial arrangements within the table layout. Existing methods predominantly treat table retrieval as a variant of text retrieval. They struggle to accurately preserve the rich structural semantics of diverse table formats during text serialization. Existing methods typically flatten tables into linear text sequences through row-wise or column-wise serialization, inadvertently discarding structural information. The problem becomes particularly acute when processing complex table layouts containing merged cells or irregular alignments, ultimately compromising retrieval performance. Moreover, existing methods struggle to handle embedded images within table cells. Notably, visual representations inherently preserve both structural and content information while being format-agnostic. This insight motivates our exploration of image-based table retrieval, as it can naturally overcome the challenges faced by existing methods. In this paper, we introduce **TaR-ViR** (Table Retrieval via Visual Representations), a new benchmark that reformulates table retrieval as a multimodal task by treating tables as images. Experiments on TaR-ViR show that this paradigm shift achieved more effective and efficient retrieval performance. Crucially, it eliminates the need for error-prone text conversion, enabling scalable collection and utilization of open-world tables. Our data are available at https://github.com/Trustworthy-Information-Access/Tab-ViR.

## 1 Introduction

Open-domain table retrieval has emerged as a critical challenge in information systems (Cafarella et al., 2008; Zhang & Balog, 2020), focusing on the accurate and efficient retrieval of structured tables from extensive corpora to satisfy diverse user information needs. As a fundamental data storage format, tables offer unique advantages for knowledge organization and dissemination due to their dual capacity for structured data storage and human-readable presentation. Their inherent spatial efficiency and logical arrangement make them indispensable for applications ranging from business intelligence to scientific research. Recent studies reveal that approximately 27% of web search queries implicitly or explicitly target tabular data (Kwiatkowski et al., 2019), highlighting the critical need for effective table retrieval systems. This substantial demand elevates table retrieval to a critical function in contemporary information systems to bridge the gap between unstructured natural language queries and structured knowledge repositories.

Table retrieval remains significantly less explored compared to conventional plain text retrieval. Existing methods predominantly treat table retrieval as a specialized variant of text retrieval (Chen

---

[†] Corresponding authors

et al., 2023). Unlike plain text, tables have their corresponding row-column structure that cannot be ignored during retrieval, suggesting that table retrieval presents unique challenges that distinguish it from standard text retrieval tasks (**?**). This is particularly the case in complex table layouts like merged cells, multi-level headers, or irregularly aligned tables. As an example, scientific datasets often use merged cells to represent logical groupings. Existing approaches focus on optimizing the understanding of table structures during the encoding process, thereby enhancing performance. Some text retrieval techniques have been effectively repurposed for table retrieval based on existing system practices. TAPAS (Herzig et al., 2020) and DTR (Herzig et al., 2021) introduce row embedding and column embedding, similar to position embedding, to represent tabular structures. UTP (Chen et al., 2023) and THYME (Li et al., 2025b) optimize the retrieval training process to enhance performance. ECAT (Li et al., 2025a) improves the performance of existing table retrievers through entity matching.

These methods have facilitated the development and application of table retrieval, but they share a limitation in practical applications. These methods inherently assume that tables are stored exclusively in textual format. However, in real-world scenarios, tables often exist in diverse formats such as spreadsheets, databases, PDFs, or embedded in web pages, rendering text-only approaches insufficient for comprehensive retrieval (Zheng et al., 2024). The limitations of text table retrievers are primarily manifested in two aspects: (1) They struggle to express the structural complexity commonly found in real-world tables, including merged cells, partitioned layouts, and hierarchical organization. Such deficiencies significantly impair the adaptability and practical utility of table retrieval systems. (2) They focus solely on textual content, disregarding multimodal components, including but not limited to visual markers and embedded images. This oversight fundamentally compromises the comprehensive understanding of tables. The text-only representation paradigm substantially restricts the generalizability and practical deployment of table retrieval systems.

Notably, visual presentation is inherently format-independent while preserving structural and content information. Given the characteristics of tables, Image-based tabular representations and multimodal retrieval methods can naturally overcome the challenges encountered by text retrievers. Unfortunately, the absence of standardized benchmarks specifically designed for visual table retrieval has significantly hindered progress in this promising research direction. To fill the gap in the field of table retrieval, we introduce **TaR-ViR** (Table Retrieval via Visual Representations), a pioneering benchmark that conceptualizes table retrieval as a multimodal problem by representing tables in visual images. Based on NQ-TABLES (Herzig et al., 2021), a text-based table retrieval dataset, we transformed it into a multimodal table retrieval benchmark. Specifically, we collect the webpage screenshots corresponding to the tables in the NQ-TABLES dataset from Wikipedia. Due to Wikipedia's dynamic and continuously updated characteristics, the crawled table images may no longer precisely match their original counterparts in the NQ-TABLES dataset. Consequently, the relevance between the queries and tables may have degraded over time. To address such issues, we designed an annotated pipeline that leverages the image understanding and reasoning capabilities of multimodal large models to assist annotators in correcting relevance annotations. Furthermore, we regenerate and check the answers corresponding to users' time-sensitive queries in TaR-ViR.

We conduct extensive experiments on TaR-ViR to compare the performance of various text retrievers and multimodal retrievers under different settings. We find that multimodal retrievers achieve competitive performance with text retrievers on table retrieval. Multimodal retrievers even outperform text retrievers in the metric of recall. These results validate the viability of image-centric table retrieval as an effective paradigm. Multimodal retrievers enable direct, large-scale retrieval of tabular data from images, eliminating the need for table extraction and data organization processes. This addresses persistent challenges in collecting and standardizing structured data across diverse domains. TaR-ViR indicates a promising paradigm of able retrieval, which can shed light on future research on this topic.

## 2 RELATED WORK

### 2.1 OPEN-DOMAIN TABLE RETRIEVAL

Open-domain table retrieval involves extracting tables from a large corpus and matching them with queries. Tables store information not only through their textual or numerical content but also through their inherent structural properties. A large amount of structurally similar or related information is

stored in tables. Due to the value of tables and their structural characteristics that differ from plain text, the research community has constructed specialized benchmarks tailored to tables to promote development in this field. NQ-TABLES (Herzig et al., 2021) OTT-QA (Chen et al., 2020) introduces table retrieval from Wikipedia to support open-domain question answering. E2E-WTQ (Pan et al., 2021) extends WikiTableQuestions into a retrieval setting and proposes Cell-Level Table Retrieval (CLTR), focusing on cell-level semantic matching. The single table sometimes fails to satisfy the user's information needs. MMQA (Wu et al., 2025) introduces Multi-Table Retrieval, which transforms multi-table retrieval tasks into multi-round single-table retrieval tasks by decomposing the multi-hop problem into a series of subproblems. Open-WikiTable (Kweon et al., 2023) expanded WikiSQL and WikiTableQuestions by incorporating a larger table corpus.

With the development of benchmarks, methods specifically designed for table retrieval have been proposed. Since pre-trained language models are primarily trained on textual corpora, they cannot comprehend structured data like tables. To improve the comprehension of tables, TAPAS (Herzig et al., 2020), TaBERT (Yin et al., 2020), and StruBERT (Trabelsi et al., 2022) improve retrieval performance by adding special neural network layers based on BERT (Devlin et al., 2019). In addition to the special network layers, training based on the table corpus improves the pre-trained model's understanding of tables (Herzig et al., 2020; Chen et al., 2023). There are also some works (Li et al., 2025a; Wang et al., 2022a; Li et al., 2025b) that demonstrate the matching characteristics of tables that differ from plain text. These methods have promoted the development of table retrieval technology, but due to the limitations of text storage, there are limitations in handling complex table structures such as merged cells, hierarchical, and nested tables. In addition to complex structures, table content may contain images that cannot be stored in text format. The limitation of text storage restricts the practical application of table retrieval in real life.

## 2.2 VISION-LANGUAGE MODELS FOR MULTIMODAL RETRIEVAL

Multimodal retrieval has long been a significant research challenge. Early works like CLIP (Radford et al., 2021), BLIP (Li et al., 2022), Align (Jia et al., 2021), SigLIP (Zhai et al., 2023), SimVLM (Wang et al., 2022b), and CoCa (Yu et al., 2022) primarily focused on learning universal representations from large-scale, weakly supervised image-text pairs. These models generally encode images and text separately, projecting them into a shared space. Most research on universal multimodal embeddings involves fine-tuning models like CLIP or BLIP, typically using simple fusion mechanisms to combine visual and language information. For instance, UniIR (Wei et al., 2023) creates multimodal embeddings by simply adding text and visual features. Furthermore, MagicLens (Zhang et al., 2024) employs shallow self-attention layers to integrate these features more effectively. These approaches have laid the groundwork for more recent multimodal large language models such as LLaVA (Liu et al., 2023) and Qwen-VL(Bai et al., 2025). Several pioneering works like VLM2VEC (Jiang et al., 2025) and GME (Zhang et al., 2025a) have demonstrated that through extensive training on large-scale, well-aligned image-text datasets, these models can learn to generate unified multimodal embeddings. The success of these methods has opened new possibilities for various downstream tasks, including retrieval, visual grounding, and so on.

## 3 BENCHMARK CONSTRUCTION

### 3.1 TASK DEFINITION

Let $D = \{(q, T^+)\}$ be a labeled dataset, where $q$ denotes a query and $T^+$ is a set of tables $\{t^+\}$ relevant to $q$. The number of elements $t^+$ in $T^+$ is different per query. A table $t$ contains an image and a textual title. The objective of table retrieval is to train a retriever that can identify relevance patterns of a query $q$ and its relevant tables $T^+$. Based on this task definition, our data construction process is illustrated in Figure 1. We employ an annotation pipeline that combines MLLMs with annotators to ensure annotation quality while improving efficiency.

### 3.2 DATA COLLECTION

Through a comprehensive review of prior work, we observe the lack of multimodal table retrieval datasets. To validate the feasibility of an image-centric table retrieval, we introduce the first specialized dataset designed for this paradigm. Existing table retrieval benchmarks can be roughly

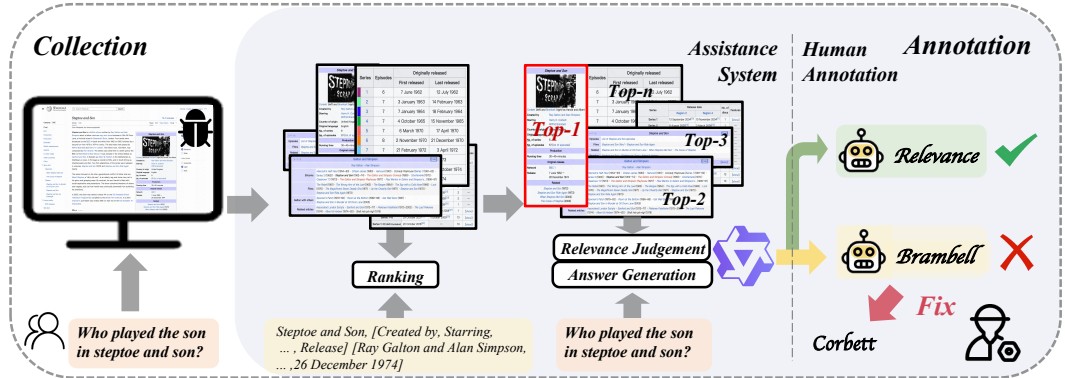

Figure 1: Construction pipeline for TaR-ViR. By combining automated annotation with an annotator, we optimize both the effectiveness and efficiency of the annotation process.

divided into two types based on their construction methodology. The first type derives from existing table question-answering (QA) benchmarks, where contextual information is removed from the original questions to generate context-independent queries, thereby repurposing the dataset for table retrieval. The second type is constructed by extracting table-related search queries from web logs, leveraging real-world user interactions to create a retrieval-oriented benchmark. The first type produces overly specific queries that poorly reflect real-world search behavior, whereas the second demands substantial annotation efforts, resulting in high development costs. However, compared to the former, it reflects the information needs of the real world and thus holds greater practical value. Therefore, we mine multimodal table retrieval datasets from web logs. Benefiting from previous work, we don't need to mine web logs from scratch. We decided to adapt a textual table retrieval dataset. Based on the NQ-TABLES (Herzig et al., 2021) dataset, we constructed TaR-ViR (Table Retrieval via Visual Representations). NQ-TABLES is a widely used benchmark for textual table retrieval, with tables collected from Wikipedia. To extend it into the multimodal domain, we systematically collect screenshots of the webpages where the tables in NQ-TABLES are embedded from Wikipedia, resulting in a collection of approximately **2 million** images.

## 3.3    ANNOTATION PIPELINE

Through systematic analysis of the crawled images, we identify two critical issues requiring resolution: redundant table occurrences and temporal shifts in query-table relevance. The redundancy issue stems from two main sources: multiple URLs referencing identical webpage content, and instances where the same table appears across different webpages. This results in significant duplication within the corpus. To address this, we employ CLIP (Chen et al., 2025) for filtering. We consider two images duplicates if their similarity score exceeds 0.9 and they share identical URL prefixes. For each set of duplicate images, we retain only a single representative instance and perform corresponding annotation consolidation to maintain data integrity. In addition, we sort the multiple images associated with a table in NQ-TABLES and pick the Top-1 result as the image representation for this table.

Given the observed temporal shifts in query-table relevance, we correct this through re-annotation. Annotation requires a large number of annotators, which is both time-consuming and costly. To address this challenge, we developed a cost-efficient

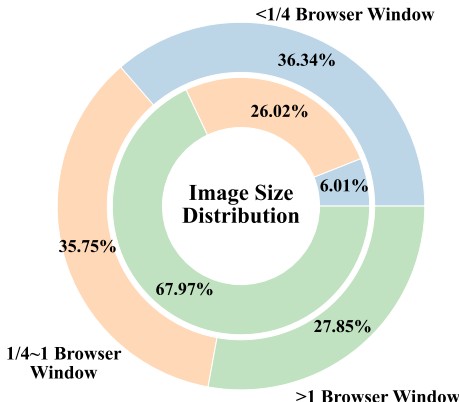

Figure 2: Distribution of table image size. The outer circle shows the distribution across the entire corpus, while the inner circle displays the corresponding tables in the test.

annotation pipeline leveraging MLLMs. Specifically, we employ Qwen2.5-VL-72B (Bai et al., 2025) for automated annotation generation, with human annotators performing a streamlined verification and correction process. The human annotation is divided into two parts: (1) **Relevance Annotation**: Annotators need to judge whether the provided pseudo-relevance results are correct. (2) **Answer Annotation**: For query-table pairs labeled as relevant by annotators, further verification is required to determine whether the answers align with the questions and reference table images. Due to the scalability constraints of our benchmark and the costs associated with human annotation, we limit manual annotation to the test set. For the training set, we retain only those instances annotated as relevant by Qwen2.5-VL-72B. We collected 1,550 query-table pairs for human evaluation, of which 1,249 ($\approx 80\%$) were validated as correct. This indicates that our automated annotation generation achieves an accuracy of approximately 80%. Although the training set was not manually refined, the automated filtering ensures its high quality.

## 3.4 DATASET STATISTICS

The statistical information of TaR-ViR are summarized in Table 1. Based on the relative size compared to the browser window, we classified tables in both the corpus and test set into three difficulty levels: **Easy** ($< 25\%$ window coverage), **Medium** (25% - 100%), and **Hard** (tables exceeding the browser window). Figure 2 shows the distribution of these categories. This classification reflects progressively challenging scenarios for MLLMs, as larger tables require more advanced visual understanding, spatial reasoning, and adaptability to increasing size and complexity.

We also provide a detailed analysis of TaR-ViR with multiple previous datasets in Table 2. We conduct a comprehensive comparison of various benchmarks by analyzing five key dimensions: dataset origin, context independence, corpus modality, data scale, and downstream tasks. Existing datasets typically originate from close-domain table question-answering datasets or web logs. Based on their provenance, we classify the context independence of different datasets into three distinct levels. In closed-domain QA benchmarks, user queries are inherently context-dependent, as they assume familiarity with the underlying tables. This type of data poses challenges for retrieval due to its abundance of directive phrases, which are often ambiguous and difficult for retrieval systems to interpret. While some datasets attempt to decontextualize queries in closed-domain QA benchmarks, they can not be representative of real-world user queries. Therefore, the user queries in this part of the dataset range from context-independent to context-dependent. Many existing datasets are derived from web user logs, which capture real-world query patterns. The user queries within them are context-independent, but such datasets are typically costly to collect and relatively small in scale. Compared to these datasets, TaR-ViR is derived from a high-quality textual table retrieval benchmark, NQ-TABLES, which reflects user information needs in real-world scenarios. Among image-centric table benchmarks, TaR-ViR is the largest in scale and simultaneously supports both retrieval and QA evaluations.

Table 1: Statistics of TaR-ViR. The table statistics are based on the images and the text obtained through OCR.

|  |  | Train | Test |
|---|---|---|---|
| Query | Count | 7,377 | 1,249 |
|  | Avg. # Tokens. | 10.47 | 10.44 |
| Table | Count | 81,839 | 81,839 |
|  | Avg. # Width. | 765.23 | 765.23 |
|  | Avg. # Height. | 1,087.35 | 1,087.35 |
|  | Avg. # Tokens. | 738.18 | 738.18 |
| # Relevant Tables |  | 1.00 | 1.02 |

Table 2: Comparison between TaR-ViR and existing datasets related to table. Modality is the storage format of the table. And # q-t Pairs refers to the number of query and table pairs contained in all sets (train and test).

|  | Source | Ctx-Independent | Modality | # q-t Pairs | # Tables | Retrieval | Q&A |
|---|---|---|---|---|---|---|---|
| WikiSQL (Zhong et al., 2017) | QA Dataset | Low | T | 80,654 | 24,241 | ✗ | ✓ |
| NQ-TABLES (Herzig et al., 2021) | Web Log | High | T | 11,628 | 169,898 | ✓ | ✓ |
| OTT-QA (Chen et al., 2020) | QA Dataset | Middle | T | 43,683 | 419,183 | ✓ | ✓ |
| E2E-WTQ (Pan et al., 2021) | QA Dataset | Middle | T | 1,216 | 2,108 | ✓ | ✓ |
| E2E-GNQ (Pan et al., 2021) | Web Log | High | T | 789 | 74,224 | ✓ | ✓ |
| StatCan (Lu et al., 2023) | Web Log | High | T | 4,468 | 1,244 | ✓ | ✓ |
| Open-WikiTable (Kweon et al., 2023) | QA Dataset | Middle | T | 67,023 | 24,680 | ✓ | ✓ |
| ComTQA (Weichao Zhao, 2024) | QA Dataset | Low | I | 9,000 | 9,000 | ✗ | ✓ |
| TabFQuAD (Faysse et al., 2025) | Web Log | High | I | 210 | 210 | ✓ | ✗ |
| **TaR-ViR** | Web Log | High | I | 8,646 | 81,839 | ✓ | ✓ |

## 4 EXPERIMENTS

### 4.1 BASELINES

To evaluate the effectiveness of different table representations (text or image), we employed two distinct types of retrievers: **Text Retrievers**, which process tabular content as plain text, and **Multimodal Retrievers**, which leverage both text and image. We compared their performance on table retrieval tasks to assess the impact of representation choice. In TaR-ViR, where tables are stored as images, we first extracted their textual content using Qwen2.5-VL-7B (Bai et al., 2025) to convert the image-based tables into structured HTML representations. These tables were then expanded row by row and transformed into sequential text format as input for the text retrievers.

### 4.2 EXPERIMENTAL SETUP

For the different baselines, we conducted training on TaR-ViR using open-source weights and frameworks*, and compared their performance differences. We implemented distinct training protocols according to model architectures: full-parameter fine-tuning was applied to BERT and CLIP-based retrievers, while LLM and MLLM-based retrievers were optimized using LoRA with a rank of 16. To ensure consistency across experiments, all models were trained for 20 epochs with a fixed batch size of 144.

### 4.3 EVALUATION METRICS

For evaluation, we employ three standard metrics: Recall, normalized discounted cumulative gain (NDCG), and mean reciprocal rank (MRR). Since the retrieved tables will be used in downstream tasks such as question answering, we use 50 as the maximum cutoff, following established practices in table retrieval (Herzig et al., 2020; Li et al., 2025a). Specifically, we report Recall@10 and Recall@50 to measure the coverage of relevant tables in top-k results. For ranking quality assessment, we adopt NDCG@5 and NDCG@10 to evaluate the graded relevance of results, accounting for the fact that different tables may contribute variably to downstream tasks. We also use MRR@5 and MRR@10 as supplementary metrics to measure how rapidly the first relevant table is retrieved, which is a crucial consideration for time-sensitive applications.

## 5 OVERALL PERFORMANCE

We extensively evaluated representative text and multimodal retrievers on the TaR-ViR. Detailed results are summarized in Table 3. Based on how data is collected and used, we categorize these retrievers into three distinct groups. By comparing the performance of retrievers within the same group and across different groups, we have summarized some interesting conclusions.

Firstly, retrievers incorporating both table titles and content exhibit superior performance, regardless of whether the input is OCR text or raw images. However, in real-world applications, table titles may not always exist. Even when present, they cannot always be extracted accurately and converted into text for storage. Considering the absence of table titles during data collection, the performance of multimodal table retrievers can reach approximately 70% of what they would achieve with complete data. When table content is presented as images, multimodal retrievers that incorporate title and content image (such as UniME and VLM2Vec) even outperform traditional OCR-based text retrievers. UniME significantly outperforms the state-of-the-art text retriever BGE on metrics such as NDCG and MRR. However, this performance advantage comes with increased model complexity. Multimodal retrievers achieving comparable performance to text-based approaches typically require substantially more parameters. Multimodal retrievers with fewer parameters struggle to integrate information from different modalities.

Comparative observations of model parameters and retrieval metrics across different retrieval approaches highlight a distinct asymmetry in how parameter scaling affects text versus multimodal retrievers. In text retrievers, BGE outperforms larger-parameter models like GTE and Qwen3-Embedding, demonstrating that model optimization, such as pretraining objectives and data quality,

---

*https://github.com/TIGER-AI-Lab/VLM2Vec

Table 3: Performance comparison of retrievers under different input modalities. R, N, and M correspond to Recall, NDCG, and MRR. **Bold** and underline indicates the optimal and suboptimal performance, respectively.

| Models | Backbones | # Params | Performance | | | | | |
|---|---|---|---|---|---|---|---|---|
| | | | R@10 | R@50 | N@5 | N@10 | M@5 | M@10 |
| *Title (Text) + Content (OCR Text)* | | | | | | | | |
| BM25 (Robertson et al., 2009) | – | – | 20.50 | 26.49 | 21.70 | 23.40 | 19.79 | 20.50 |
| BIBERT (Devlin et al., 2019) | BERT-Base | 109M | 89.08 | 95.64 | 68.82 | 70.67 | 63.92 | 64.71 |
| GTE (Li et al., 2023) | Qwen2 | 1.54B | 87.46 | 96.84 | 64.68 | 67.58 | 59.99 | 61.68 |
| BGE (Chen et al., 2024) | BERT-Base | 109M | 92.15 | 98.23 | 72.82 | 74.69 | 68.18 | 68.95 |
| Qwen3-Embedding (Zhang et al., 2025b) | Qwen3 | 4.05B | 89.30 | 98.15 | 71.93 | 73.83 | 68.04 | 68.84 |
| *Title (Text) + Content (Image)* | | | | | | | | |
| CLIP (Radford et al., 2021) | CLIP-Large | 428M | 36.07 | 51.69 | 26.15 | 27.47 | 24.14 | 24.69 |
| SigLIP (Zhai et al., 2023) | SOViT-400m | 878M | 48.61 | 63.53 | 33.43 | 35.89 | 30.74 | 31.75 |
| GME (Zhang et al., 2025a) | Qwen2-VL | 2.21B | 59.80 | 68.80 | 43.68 | 45.84 | 40.15 | 41.06 |
| UniSE-MLLM (Liu et al., 2025) | Qwen2-VL | 2.21B | 92.23 | 98.53 | 68.01 | 70.18 | 62.17 | 63.09 |
| VLM2Vec (Jiang et al., 2025) | Qwen2-VL | 2.21B | 90.69 | 97.53 | 67.89 | 70.33 | 62.72 | 63.68 |
| VLM2Vec (Jiang et al., 2025) | Qwen2-VL | 7.07B | **94.23** | **98.92** | 70.56 | 72.81 | 64.80 | 65.76 |
| UniME (Gu et al., 2025) | LLaVA-1.6 | 7.57B | 93.38 | 98.76 | **75.62** | **77.39** | **71.44** | **72.20** |
| *Content (Image)* | | | | | | | | |
| DSE (Ma et al., 2024) | Phi-3-V | 4.15B | 66.45 | 77.61 | 51.44 | 53.76 | 48.59 | 49.54 |
| ColPali(Faysse et al., 2025) | Qwen2-VL | 2.21B | 63.59 | 72.89 | 48.93 | 51.15 | 46.15 | 47.05 |

is more critical than parameter scale in text embedding. However, in multimodal retrievers, results demonstrate a consistent improvement in performance with increasing model parameters and capabilities. This finding indicates that joint visual-text representation learning requires models with strong foundational capabilities, making larger parameter scales indispensable.

The above results reveal two paradigms for addressing multimodal table retrieval. One utilizes MLLMs to extract text from images, transforming the multimodal table retrieval task into a pure text-based table retrieval task. Relevant tables are then retrieved through a small text retriever like BGE. The other directly employs an MLLM-based multimodal retriever to retrieve the relevant tables. Compared to text retrievers, multimodal retrievers do not require a preprocessing step to convert images into text to be input into the retrievers. Multimodal retrievers can directly process text and images with simplicity and efficiency. The direct utilization of both text and images offers a streamlined and effective approach. However, text retrievers based on OCR content are more flexible in use. The OCR content of the table can be obtained by calling MLLMs via the API. In practical applications, choosing between these two methods requires balancing resource and efficiency.

## 6 FURTHER ANALYSIS

### 6.1 IMPACT OF DIFFERENT TABLE FORMAT

Multimodal retrievers have demonstrated competitive performance in table retrieval. However, because of differences in the training data used and optimization procedures across retrieval systems, the performance comparisons in Table 3 are not entirely fair. To eliminate performance discrepancies arising from different training processes, we independently trained dedicated text retrievers and multimodal retrievers using Qwen2-

Table 4: Performance comparison of retrievers with different input formats. All retrievers are implemented using Qwen2-VL-2B. The only difference lies in the format.

| Input Format | R@10 | N@5 | M@5 |
|---|---|---|---|
| *Title (Text) + Content (OCR)* | 89.00 | 65.91 | 61.46 |
| *Title (Text) + Content (Web)* | 90.30 | 65.20 | 60.64 |
| *Title (Text) + Content (Image)* | **91.69** | **67.42** | **62.19** |
| *Title (Image) + Content (Image)* | 82.30 | 58.09 | 53.32 |

VL-2B as the base architecture. For rigorous experimental control, we maintained identical hyperparameter configurations across all models, modifying only the input data format to accommodate different modalities. The result is shown in the Table 4. MLLMs can parse table information directly from images while learning the relevance between queries and tables. We also observed variations between different input contents. When using the original table content from web pages instead of OCR results as input for the text retriever, it showed different trends across different metrics. The

metric of recall has increased, while NDCG and MRR have shown a downward trend. Through case observation, we have identified the causes behind this phenomenon. For retrieval, more input information does not necessarily yield better results. Given the presentation of tables in images, MLLMs demonstrate a systematic preference for extracting information that has been visually emphasized through color contrast, font variations, and other typographic cues. This behavior mirrors human information-seeking patterns. As a result, OCR-based input leads to superior performance in ranking-related metrics, including NDCG and MRR.

## 6.2 IMPACT OF INFORMATION VOLUME ON TABLE RETRIEVERS

Tables can store large amounts of data through a simple row and column structure. In practice, tables vary in size from single cells to multi-page layouts. To investigate the performance differences of tables with varying information under different retrievers, we divided the test set into three groups based on the size of the corresponding relevant tables for each query, following the classification standards outlined in Section 3.4. The results are shown in Table 5.

Table 5: Performance comparison for tables with different information volumes. The best performance in each group is denoted by $\star$, $\dagger$, and $\diamond$, respectively.

| | Title | Content | Metrics | | | | | |
| | | | R@5 | R@10 | N@5 | N@10 | M@5 | M@10 |
|---|---|---|---|---|---|---|---|---|
| **Easy** | Text | OCR Text | 87.65$\star$ | 96.29$\star$ | 71.86$\star$ | 74.39$\star$ | 67.07$\star$ | 67.80$\star$ |
| | Text | Web Text | 81.81 | 92.20 | 68.19 | 71.33 | 63.42 | 64.71 |
| | Text | Image | 85.71 | 92.20 | 68.85 | 70.82 | 63.20 | 63.87 |
| **Medium** | Text | OCR Text | 84.59 | 91.23 | 72.66 | 74.80$\dagger$ | 68.66$\dagger$ | 69.53$\dagger$ |
| | Text | Web Text | 88.07 | 92.66 | 71.73 | 73.24 | 66.18 | 66.81 |
| | Text | Image | 90.21$\dagger$ | 92.96$\dagger$ | 73.84$\dagger$ | 74.75 | 68.32 | 68.69 |
| **Hard** | Text | OCR Text | 76.12 | 87.50 | 62.82 | 66.51 | 58.23 | 59.77 |
| | Text | Web Text | 80.93$\diamond$ | 89.65 | 62.92 | 65.77 | 57.07 | 58.20 |
| | Text | Image | 80.81 | 91.51$\diamond$ | 65.29$\diamond$ | 68.76$\diamond$ | 60.04$\diamond$ | 61.53$\diamond$ |

When tables are small, text retrievers using OCR or original text content from the webpage perform better than multimodal retrievers. However, as tables grow in both structural complexity and physical dimensions, MLLMs struggle to accurately extract and parse all textual content appearing in tabular images. The image-centric table retrieval paradigm is gradually demonstrating its advantages. The multimodal approach demonstrates particular effectiveness in preserving the complete informational integrity of tabular images, thereby enabling more precise query-table matching. Comparison of results using OCR text and original text from the webpage, we observed the same phenomenon in Section 6.1: OCR-based text retrievers outperformed the original text in ranking-related metrics. However, as the table information in the image increases, this advantage gradually diminishes. When a table contains information spanning more than one browser window, information highlighted through visual cues becomes difficult for MLLMs to accurately extract, leading to a decline in retrieval performance.

## 7 RAG APPLICATION

The advent of large language models has transformed retrieval systems into auxiliary knowledge providers that primarily supply input to LLMs, rather than operating as standalone user interfaces. This functional transition is most apparent in applications such as question answering and text summarization systems. The TaR-ViR we proposed not only includes annotations for relevance but also annotations for the answers to user questions given the relevant tables. In this section, we demonstrate the evaluation of the RAG system to highlight the value of TaR-ViR. For retrievers, we employ a text retriever and a multimodal retriever trained on Qwen2-VL-2B. For generators, we evaluate both LLM and MLLM under two distinct retriever configurations. These diverse combinations can illustrate the modal preferences of RAG systems when processing tables.

The experimental results, obtained using Mistral (Jiang et al., 2023), Llama3 (Grattafiori et al., 2024), Qwen3 (Yang et al., 2025), LLaVA-OneVision (Li et al., 2024), and

Qwen2.5-VL (Bai et al., 2025) are presented in Table 6. Our data is based on NQ-TABLES, a dataset comprising a large-scale collection of factual questions. Following standard practice in question-answering evaluation, we measure accuracy by verifying whether the annotated ground-truth answers appear in the models' generated responses. Table 4 demonstrates the superior performance of the multimodal retriever (trained on Qwen2-VL-2B) compared to its text-based counterpart. Similar conclusions were also demonstrated in downstream QA evaluations. Qwen2.5-VL achieved better performance when tables were input as images. However, overall, text-based LLMs exhibit superior upper-bound performance. To answer questions corresponding to image-based tables, MLLMs must simultaneously possess precise visual parsing, structured semantic understanding, and symbolic reasoning capabilities. Current MLLMs, however, have not yet achieved complete integration of these essential competencies. This also reflects the potential value of TaR-ViR from another perspective. It demonstrates the limitations of current MLLMs.

Table 6: Performance evaluation of the RAG system across different configurations. Gray indicates multimodal input.

| Retriever | Generator | Accuracy | | |
|---|---|---|---|---|
| | | n=1 | n=3 | n=5 |
| Qwen2-VL-2B (*Text Retrieval*) | Mistral-7B | 31.43 | 36.15 | 35.92 |
| | Llama3.1-8B | 31.27 | 35.17 | 37.21 |
| | Qwen2.5-VL-7B | 33.46 | 37.78 | 38.51 |
| | Qwen3-8B | **46.57** | 55.35 | 58.38 |
| | LLaVA-OneVision | 21.25 | 20.84 | 18.21 |
| | Qwen2.5-VL-7B | 34.85 | 40.24 | 40.87 |
| Qwen2-VL-2B (*MultiModal Retrieval*) | Mistral-7B | 30.61 | 36.56 | 36.97 |
| | Llama3.1-8B | 30.86 | 35.83 | 35.66 |
| | Qwen2.5-VL-7B | 31.59 | 38.02 | 39.33 |
| | Qwen3-8B | 44.05 | **55.86** | **58.55** |
| | LLaVA-OneVision | 19.62 | 20.68 | 20.35 |
| | Qwen2.5-VL-7B | 32.82 | 41.10 | 42.46 |

## 8 CASE STUDY

To visually analyze how image-centric table retrieval paradigms impact performance, we present a concrete case study in Figure 3. The figure compares the results of the same query processed through both text-only and multimodal retrieval systems. Notably, while the textual content of the table image was accurately extracted via OCR, the multimodal retriever achieved superior ranking performance by effectively leveraging visual-semantic features. This improvement can be attributed to the additional visual cues—such as photographs, colors, and layout—embedded in the image. These visual elements emphasize key information of the table, enhancing the multimodal retriever's ability to identify and prioritize specific content.

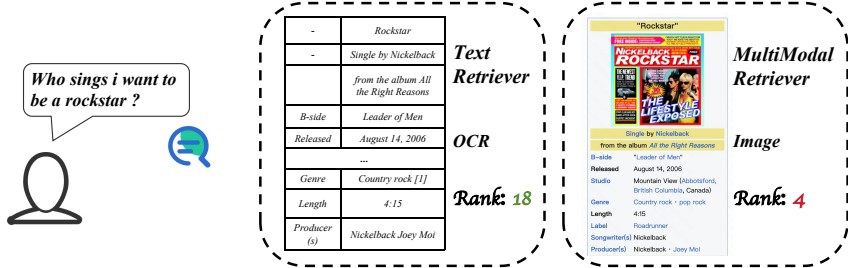

Figure 3: Retrieval result comparison between different storage formats. The text retriever and multimodal retriever are Qwen2-VL-2B trained using OCR and multimodal data.

## 9 CONCLUSION

In this work, we propose TaR-ViR, a multimodal table retrieval benchmark, and demonstrate the effectiveness of multimodal retrievers in table retrieval. Our experiments show that multimodal retrievers achieve competitive performance compared to text retrievers on this task. Multimodal retrievers can directly analyze table content within images while learning the semantic relevance between the table's visual representation and user queries. Furthermore, unlike text retrievers, multimodal retrievers can directly process images without converting them into text, offering significant efficiency advantages. This result also demonstrates that adopting a more flexible storage method, like images, can effectively organize and manage the massive amounts of tabular data encountered in real-world applications, thereby better addressing users' information needs.

## ETHICS STATEMENT

TaR-ViR we propose is derived from existing open-source datasets. Additionally, the data we have scraped is based on publicly available content from Wikipedia and does not involve any privacy information belonging to organizations or individuals. We paid the annotators their corresponding compensation for their work in constructing the dataset.

## REPRODUCIBILITY STATEMENT

We propose a novel multimodal table retrieval dataset TaR-ViR, leveraging publicly accessible Wikipedia data throughout the process. Furthermore, our training process is based on open-source model weights and code implementations. Our data will also be made open-source to the public, promoting the advancement of the field of table retrieval.

## ACKNOWLEDGEMENT

This work was funded by the National Natural Science Foundation of China (NSFC) under Grants No. 62302486 and No. 62441229, the Innovation Project of ICT CAS under Grants No. E361140, the CAS Special Research Assistant Funding Project, and the project under Grants No. JCKY2022130C039.

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

## A   THE USE OF LARGE LANGUAGE MODELS

In this paper, we employed a large language model (LLM) in two auxiliary capacities: (1) to assist with wording refinement, and (2) to generate data visualization code templates. Importantly, all fundamental concepts, theoretical frameworks, experimental methodology, and the complete initial manuscript were independently developed and authored by the authors. All LLM-generated content was manually validated by the authors.

## B   PROMPT FOR RAG EVALUATION

**Answer Generation:**
You need to answer the user's questions based on the tables provided to you.
**[User Question]**: *<Question>*,
Below is the information from *<Table Count>* tables.
[Table 1]: *<Title>*:*<Content>*.
[Table 2]: *<Title>*:*<Content>*.
...
**[Requirements]**: You must output the answer directly. Outputting anything other than the answer is not permitted.
**[Output]**:

## C   PROMPT FOR TABLE OCR

**Table Content Extraction:**
You are an OCR table recognition expert with years of experience.
Goals: You need to recognize the content in the given image and output the results in a CSV table format.
Constraints:
1. You need to recognize the content in the image and completely extract the content from each table cell, placing it into a CSV table structure.
2. There may be placeholders in the table cells in the image, such as "-", "—", "/", etc., which need to be recognized.
3. The output table structure must follow the structure in the image exactly, with complete consistency.
4. Pay special attention to merged cells in the image to ensure the structure is correct.
5. For images with a lot of content, ensure the output is complete, without omitting or fabricating any information.
6. The final output needs to be in CSV table format.
7. Extract the main content of the table and limit the content to within 2000 tokens.
Initialization:
Please think carefully and output the CSV table result.
[Table ]: *<Title>*:*<Image>*.
**[Output]**:

