# OpenReview forum: "Beyond Text-Only: Towards Multimodal Table Retrieval in Open-World"
_ICLR.cc/2026/Conference — ICLR 2026 Poster_

### Official Review · Reviewer_EvmC · 2025-10-30

**Soundness:** 2
**Presentation:** 2
**Contribution:** 2
**Rating:** 4
**Confidence:** 4

**Summary:**

This paper proposes TaR-ViR, a new benchmark that reframes open-domain table retrieval as a multimodal task by treating tables as images rather than serialized text. The authors crawl Wikipedia screenshots for tables and derive a corpus with 81,839 tables and 8,646 query–table pairs. They use a hybrid MLLM-assisted annotation pipeline: using Qwen2.5-VL-72B proposes labels which are then partially verified by humans. The evaluation of the paper compares text retrievers with multimodal retrievers, and reports that large multimodal retrievers can match or surpass strong text retrievers on several ranking metrics. They also include a RAG study showing modest gains when using multimodal retrievers for table-QA inputs, while text-only LLMs still achieve a higher upper bound overall.

**Strengths:**

**1. Well-motivated**: Directly operating on table images addresses notorious issues with serialization losses, such as merged cells, hierarchical headers, embedded figures that harms text-only pipelines.

**2. Scale & coverage**: The pipeline begins from 2M screenshots and yields a large image-centric retrieval benchmark with 81,839 tables; clear dataset statistics are provided.

**3. Practical annotation design**: The MLLM-assisted relevance and answering workflow with human verification is cost-aware and reports empirical quality (80% held rate), with manual curation reserved for the test set.

**4. Evidence of benefits**: On TaR-ViR, strong multimodal models (e.g., VLM2Vec-7B) can outperform the best text retrievers (e.g., BGE) on ranking metrics when using title+content images.

**Weaknesses:**

**1. Benchmark scope**: The corpus is Wikipedia centric. Table images, such as scanned documents, enterprise spreadsheets, and PDFs beyond Wikipedia layouts is not evaluated, limiting external validity.

**2. Novelty**: The novelty of hybrid data collection pipeline of TaR-ViR is limited.

**3. RAG gains**: The RAG table shows modest benefits from multimodal retrieval, while text LLMs still dominate, leaving the practical payoff ambiguous for QA pipelines that can rely on OCR and text LLMs.

**4. Annotation noise**: The auto-labeled training set is reported with 80% precision suggests non-trivial label noise that could bias model training. More rigorous robustness analysis, such as different MLLMs' label-noise sensitivity, would strengthen claims.

**5. Reliance on titles**: Several best results assume title and image are both available. In the wild, titles may be missing or noisy. Extra results on how performance degrades when titles are unavailable are helpful.

**Questions:**

Check the weaknesses.

---

> ### Author Response · Authors · 2025-11-24
>
> Thank you for your valuable feedback on our work. We will respond to each of your comments one by one.
>
> For Weekness 1:
>
> Tables can be used in a wide variety of scenarios. TaR-ViR should incorporate tables from as many different scenarios as possible. However, in the construction process, due to commercial constraints, we cannot build a substantial corpus collection based on corporate spreadsheets and documents. Moreover, we have a limited number of human annotators. Therefore, the use of Wikipedia as a source for the table corpus is primarily driven by considerations of cost and scale. We have leveraged the relevance annotations from NQ-TABLES, which will reduce the cost of secondary annotation.  Data scale is essential for retrieval. In terms of scale, the number of tables contained in TaR-ViR far exceeds that of datasets constructed manually from documents and PDFs, as shown in Table 2. For tables within PDFs or documents, we plan to conduct long-term collection and relevance annotation to advance the development of related evaluations.
>
> For Weekness 2:
>
> We drew upon existing methods for synthetic datasets to propose the TaR-ViR construction pipeline. The pipeline itself is not our core contribution. We aim to validate the feasibility of a multimodal table retrieval paradigm centered on images and determine whether it can replace text-based retrieval methods. However, there is currently a lack of available large-scale datasets. Therefore, we hope to develop an automated method that can be constructed at a low cost. Although our construction pipeline is simple, it is effective.
>
> For Weekness 3:
>
> In Table 3, we demonstrate the advantages of reconstructing the table retrieval task through the multimodal paradigm. Furthermore, we would like to analyze the impact of such advantages on downstream tasks like QA. Through evaluations in RAG scenarios, we found that multimodal retrieval-based RAG performs better when the input context contains multiple tables, regardless of whether LLMs or MLLMs are used in QA. This indirectly demonstrates that multimodal retrieval can yield performance gains for downstream tasks. However, the performance of QA systems using images as input is relatively poor. This highlights the current limitations of MLLMs. We thought that the training process for current MLLMs primarily focuses on natural images, resulting in a weaker understanding of non-natural images. This may represent an important area for MLLM optimization.
>
> For Weekness 4:
>
> Let me elaborate on 80% precision. During human annotation, we categorize these negative tables into two types: one involves low-quality images collected, and the other pertains to content that is thematically relevant but contextually irrelevant. At this stage, we employed the Qwen2.5-VL 72B model to complete this process. Due to cost and time constraints, we did not conduct robustness analysis using results from multiple different MLLMs. As you suggested, we are currently using different MLLMs to generate annotations and analyze the corresponding annotation quality. We will update the subsequent results in the paper.
>
> For Weekness 5:
>
> Thank you for your valuable comments. We have supplemented the experiments to demonstrate the importance of the content in the table. We use VLM2Vec to compare performance across different input contents. We can observe that when the title is unavailable, the performance shows a significant decline. However, encoders such as DSE, designed specifically for visual documents, and multi-vector encoders like ColPali can produce superior results.
>
> |                   | Model   | Backbone | R@10 | N@5  | M@5  |
> |-------------------|---------|----------|------|------|------|
> |  **With title**   | VLM2Vec | Qwen2-VL | 90.69| 67.89| 62.72|
> | **Without title** | VLM2Vec | Qwen2-VL | 64.76| 47.90| 45.02|
> | **Without title** | ColPali | Qwen2-VL | 63.59| 48.93| 46.15|
> | **Without title** | DSE     | Phi-3-V  | 66.45| 51.44| 48.59|

---

> > ### Author Response · Authors · 2025-11-28
> >
> > Dear Reviewer EvmC,
> >
> > As we approach the end of the discussion period, we kindly invite you to share any additional thoughts regarding our response to your concerns above. We sincerely appreciate your efforts and valuable feedback thus far. You may have unintentionally overlooked some of the experimental results earlier, and we have provided clarification in the rebuttal. In addition, we have included the ablation studies you requested.
> >
> > We are looking forward to your response!
> >
> > Best regards,
> >
> > The Authors

---

### Official Review · Reviewer_LgPw · 2025-10-30

**Soundness:** 3
**Presentation:** 3
**Contribution:** 3
**Rating:** 8
**Confidence:** 4

**Summary:**

This paper introduces TaR-ViR, a vision-based benchmark for table retrieval, providing an in-depth analysis of the fundamental limitations of text-based table retrieval methods when handling complex table structures and embedded images. It also proposes a novel paradigm that treats tables as images for retrieval.
The benchmark is constructed with a rigorous methodology that balances both quantity and diversity, and offers valuable insights for future research through comprehensive experiments.

**Strengths:**

1. This work provides an in-depth summary of the fundamental limitations of text-modal tables in handling complex table structures and embedded images.
2. The benchmark construction methodology is sound, cost-effective, and exhibits sufficient volume and diversity.
3. The experiments are reasonably comprehensive and offer a thorough comparison of various retrieval configurations.

**Weaknesses:**

1. Some expressions are overly strong. For example, the text modality is not entirely incapable of representing certain heterogeneous tables, and it must be admitted that text-based retrieval can be simple and fast (especially when the original data is already in text form). Similarly, table-modal tables are not perfect either — for instance, visual modalities also face challenges when dealing with very large database tables.
2. RAG also includes reranking. Although more combinations would significantly increase experimental complexity, it is still recommended to add some related experiments.
3. Are the criteria for difficulty classification somewhat oversimplified?

**Questions:**

1. As a benchmark for RAG, it is recommended to incorporate more table-specific performance metrics, such as robustness to row/column permutation (for permutable tables), rendering method robustness, and image resolution robustness.
2. The data deduplication phase employed the CLIP model, while CLIP is also used in subsequent comparisons. This may introduce potential bias.
3. This paper identifies limitations of text-modal tables, such as difficulty in handling visual elements (e.g., national flags, emojis). However, the benchmark statistics do not specify the proportion of such tables, making the benchmark less targeted.

---

> ### Author Response · Authors · 2025-11-24
>
> In response to your valuable feedback, we have provided a detailed reply to each of your comments below.
>
> For Weakness 1:
>
> Tables, as a complex format of data representation, can be formalized in different ways according to various task scenarios. Thank you for pointing out the weaknesses in our presentation. We will make the corrections in the next version. We agree with you very much that text is a simple and effective method for constructing table inputs. In table retrieval scenarios, text retrievers demonstrate competitive performance. We explored whether multimodal approaches could improve this task performance.  Compared to textual representations, storing tables as images facilitates data collection and enables rapid construction of domain-specific table retrievers. Whether in text or multimodal formats, there remains a lack of capability for processing large tables. This requires further exploration to resolve.
>
> For Weakness 2:
>
> Thank you for your insightful comment. In this paper, we demonstrate the advantages of utilizing the multimodal paradigm to address table retrieval. We want to analyze whether such advantages will impact downstream QA tasks. Therefore, we compared results using a simple RAG strategy rather than a well-designed complex pipeline. We found that when the context contains multiple tables, QA results based on multimodal retrievers perform better, regardless of whether downstream tasks select LLMs or MLLMs to generate responses.
>
> For Weakness 3 & Question 1 & Question 3:
>
> Our data collection process was achieved by taking screenshots directly from Wikipedia pages. Due to limitations in image recognition and editing algorithms, we lack the appropriate tools to analyze and edit screenshots of tables. As you mentioned, table data itself has special characteristics that distinguish it from plain text. We are attempting to re-collect data while preserving comprehensive original information for diverse analysis and evaluation.
>
> For Question 2:
>
> We used CLIP to filter the original collected data. Our filtering is list-wise; for each table in NQ-TABLES, we filter multiple table images crawled from its stored webpage. We agree with your suggestion. In subsequent training, we have selected CLIP as one of our evaluation baselines. This may introduce potential bias. However, since CLIP underwent further training on the TaR-ViR training set, its training aimed to fit the corpus-wise relevance distribution. Therefore, we thought this bias was not significant, and CLIP still learned the relevance in TaR-ViR. Its performance lacks competitiveness among the baselines. Additionally, we simultaneously demonstrated the performance of SigLIP on TaR-ViR. The results from both models collectively represent the performance of small models on the TaR-ViR.

---

### Official Review · Reviewer_U5D7 · 2025-11-01

**Soundness:** 3
**Presentation:** 4
**Contribution:** 2
**Rating:** 6
**Confidence:** 4

**Summary:**

The authors extend the existing open-domain table retrieval paradigm, framing it as a visual retrieval task rather than the previous text retrieval task. Based on this, they construct TaR-ViR, the first multimodal table dataset that supports both QA and retrieval.

**Strengths:**

The authors present their ideas clearly with fluent logic, and the overall experiments are comprehensive.
1. Novel Paradigm for Table Retrieval: The author fills a critical gap in existing research by being the first specialized benchmark that reformulates table retrieval as a multimodal task, treating tables as images instead of relying solely on text. Experiment results demonstrating that image-centric table retrieval can outperform text-based methods (especially in recall and handling complex structures) while eliminating error-prone OCR/text conversion, the work proposes a more efficient and flexible paradigm for real-world table retrieval. It also supports both retrieval and QA evaluations, enhancing its utility for diverse downstream tasks.
2. High-Quality Data Construction: Leveraging Wikipedia’s rich table resources and NQ-TABLES’ foundation, the dataset ensures real-world relevance while resolving key issues (redundancy, temporal relevance shifts) via a cost-efficient annotation pipeline combining MLLMs (Qwen2.5-VL-72B) and human verification.

**Weaknesses:**

1. During the data construction process, the authors initially collected 2 million table screenshots based on NQ-TABLES, which contains approximately 100,000 document data entries. Why is there such a large discrepancy in the data collection process? It is necessary to specify how the data volume changes at each subsequent step of data processing, as well as the data splitting method and whether it is consistent with that of NQ-TABLES.
2. The authors also mention in the paper that the comparisons in the experiments of Section 5 are unfair, and I agree with this view. I believe the conclusions derived here are not sufficiently compelling—they only demonstrate that table retrieval tasks can be completed using visual elements instead of text. Additionally, I am concerned about the results of "title + content (web)", which the authors have not provided here.
3. In the experiments of Section 6, the authors observe that performance degrades due to OCR limitations when tables become complex, but performs better on simple tables. I am curious whether combining image and text content would yield better results; furthermore, regarding the definition of complex tables, I believe relying solely on size is insufficient, and the authors could attempt to derive tables of different complexities by decomposing table parsing structures.
4. In Table 6, the authors should present the improvements of multimodal retrieval compared to text retrieval. Currently, for RAG applications, it does not show significant advantages over text retrieval, which raises doubts about whether this paradigm is superior to text-based approaches. Moreover, the performance disadvantage should not be solely attributed to MLLMs.
5. The authors also state that there are images in tables that existing methods cannot handle. If MLLMs are utilized to embed images into text sequences, what would the overall performance be?

**Questions:**

See weakness part

---

> ### Author Response · Authors · 2025-11-24
>
> Thanks for your detailed feedback. We've provided a response to each point below.
>
> For Weakness 1:
>
> Let me explain the reason for the change in the number of tables. In NQ-TABLES, each table contains the URL of the website it is embedded in. To prevent links from becoming inactive, NQ-TABLES provides more than one reference webpage for some tables. At the same time, a Wikipedia page contains a large amount of content wrapped in <table> </table> tags, which we treat as tables.  Therefore, we can extract multiple table images from a single webpage. Among the numerous tables collected, the majority are redundant. We use matching to filter the collected images. We utilized CLIP to filter these images. For the division of training and test sets, we maintain the same division as NQ-TABLES. Thank you for your suggestion. We will provide further clarification and explanation in the main text.
>
> For Weakness 2:
>
> Thank you for pointing out the shortcomings in our expression. Considering that we employed a Qwen2.5-VL-7B model to extract and organize tabular information from images, text retrieval incurs additional processing overhead compared to multimodal retrievers. Therefore, through Table 3, we aim to demonstrate that: First, table retrieval tasks can be accomplished using visual elements instead of text. Second, under the same computational overhead, multimodal retrievers outperform text-based retrievers. For a fair comparison, we can analyze the results of GTE and VLM2Vec (Qwen2-VL-2B), which represent adaptations and improvements of the same model for text retrieval and multimodal retrieval, with the latter demonstrating superior performance. Regarding “title + content (web)”, we need to elaborate further. We aim to analyze whether MLLMs can better learn table structures from HTML by varying the organization of the tables. However, in the analysis presented in Table 4, we found that LLMs can learn the structure of tables from the data without requiring additional operations. Therefore, we show the results of “title + content (OCR)” in a unified manner.
>
> For Weakness 3:
>
> This is a very interesting setting, and we have supplemented the relevant results. We found that supplementing image information with OCR content did not improve performance. When both image and OCR inputs are present simultaneously, the input length becomes substantial, placing greater demands on the capabilities of multimodal retrievers. Additionally, MLLMs need to learn one-to-one correspondences between textual content and visual content, which may pose challenges for MLLMs in understanding tabular data.
>
> |  Input                 | Model   | Backbone | R@10 | N@5  | M@5  |
> |-------------------|---------|----------|------|------|------|
> |  title + content (Image)   | VLM2Vec | Qwen2-VL | 90.69| 67.89| 62.72|
> |  title + content(Image) + content (OCR)   | VLM2Vec | Qwen2-VL | 90.91 | 65.65| 59.93|
>
> Regarding the issue of classifying table complexity, we agree with your opinion. During data collection, we saved screenshots of the tables within the webpages. This leads us to rely on the image processors for further parsing when analyzing tables. However, we've found that existing parsing tools cannot accurately extract table information, especially when dealing with tables containing substantial content and limited resolution. Therefore, we categorized the collected tables based on their size.
>
> For Weakness 4:
>
> In Table 3, we highlight the benefits of reframing table retrieval within a multimodal framework. We further explore how these advantages influence downstream tasks such as question answering (QA). Evaluations in retrieval-augmented generation (RAG) settings reveal that multimodal retrieval-based RAG consistently outperforms other approaches when the input context includes multiple tables—regardless of whether LLMs or MLLMs are employed for QA. This indirectly underscores the positive impact of multimodal retrieval on downstream task performance. That said, we observed that QA systems using images as input tend to perform poorly, which points to current limitations in MLLMs. Considering the broad applicability of images, we will explore training MLLMs specifically for table images in the future.
>
> For Weakness 5:
>
> Thank you for your comment. We consider this a valuable suggestion. During the process of collecting data from Wikipedia, we only saved full-page screenshots of images and did not retain images appearing within table cells. Collecting the table data is time-consuming, making it difficult for us to recompile the data during the discussion. Therefore, we cannot provide the experimental results. However, we thought embedding image encoding into the text sequence might yield higher performance, leveraging the capabilities of MLLM while preserving as much information as possible. This requires data collection to meet corresponding specifications, while also demanding that MLLMs possess the capability to process long-context information.

---

### Official Review · Reviewer_5rq1 · 2025-11-04

**Soundness:** 2
**Presentation:** 3
**Contribution:** 2
**Rating:** 4
**Confidence:** 4

**Summary:**

This paper introduces TaR-ViR, a multimodal benchmark that redefines open-domain table retrieval by treating tables as images rather than text sequences. The work argues that text-only approachesfail to capture the rich structural and spatial semantics of real-world tables. TaR-ViR extends the NQ-TABLES dataset by collecting approximately 2 million table screenshots from Wikipedia and aligning them with natural-language queries via a semi-automated annotation pipeline that leverages MLLMs.
Comprehensive experiments compare text-based retrievers against multimodal ones, showing that multimodal retrievers achieve competitive or superior performance, particularly in recall and large-scale retrieval efficiency.

**Strengths:**

- The benchmark scale is large.

- TaR-ViR provides a full framework integrating visual annotation, OCR-based comparison, and RAG-based downstream tasks.

**Weaknesses:**

- Although the paper claims open-world applicability, TaR-ViR relies entirely on Wikipedia-sourced tables. These are relatively clean, consistently formatted, and visually homogeneous.

- The reliance on Qwen-VL for pseudo-labeling and 80% correctness in human verification introduces potential label noise. The test set’s limited manual validation raises concerns about bias propagation, especially since multimodal retrievers were trained on partially machine-labeled data.

- While Section 7 integrates TaR-ViR into a RAG QA pipeline, the downstream results show only marginal accuracy improvements and rely primarily on recall.

- The paper’s primary contribution is dataset construction and evaluation rather than a new retrieval architecture. Its impact may hinge on community adoption rather than algorithmic innovation.

**Questions:**

- Would performance degrade if the tables included handwritten or low-quality scanned data, given that all current images are digital screenshots?

- Could multimodal retrievers trained on TaR-ViR generalize to document-level retrieval tasks where tables coexist with charts or paragraphs?

- The benchmark focuses on retrieval efficiency but omits latency and compute cost comparisons between OCR-based and pure visual retriever?

---

> ### Author Response · Authors · 2025-11-24
>
> We appreciate your feedback and will respond to each comment point by point.
>
> For Weakness 1:
>
> Although our tables are collected from Wikipedia, they are not consistent in format and visual presentation. During collection, we treated the content enclosed between <table></table> tags as tables. HTML tags offer rich expressive capabilities. Within Wikipedia pages, elements like Infoboxes and navigation bars are organized as tables. Even within the main body of a single webpage, tables appear in various formats. During manual annotation, we observed a wide variety of tabular presentation styles. Tables included in TaR-ViR exhibit diversity rather than a monotonically consistent structure.
>
> For Weakness 2:
>
> Data annotation for table retrieval is highly labor-intensive, creating a trade-off between dataset scale and precision. TaR-ViR aims to build a dataset oriented toward real-world application scenarios, thus requiring a large scale of data. This necessitates the efficient utilization of annotations. We adopted the following strategy to achieve a balance of scale and precision: for the test set, we conducted comprehensive rather than partial annotation to ensure its accuracy. Although MLLMs can produce high-quality annotations, we acknowledge that they cannot fully replace human effort. We will explore how to efficiently combine human annotation with MLLM automatic annotation in future work, iteratively improving the quality of the benchmark. This requires a long-term investment of time and resources.
>
> For Weakness 3:
>
> We validated the advantages of MLLMs in table retrieval in Table 3. Image-centric retrieval paradigms achieve better performance than OCR text-centric retrieval paradigms. In Section 7, we aim to verify whether such discrepancies persist during the QA phase. We separately trained Qwen2-VL-2B as a text retriever and a multimodal retriever to retrieve tables based on queries, providing input for LLMs and MLLMs. We found that when the input context contains multiple tables, RAG based on multimodal retrieval works better. This demonstrates the advantages of multimodal patterns for table retrieval.  During the QA phase, MLLMs demonstrated relatively poor performance. We thought this was because table images are not natural images and account for a small proportion in the training process of MLLMs. Therefore, when handling table-related input, its ability is poor. This may represent a potential optimization direction for MLLMs.
>
> For Weakness 4:
>
> In this paper, we primarily propose a new paradigm for image-centric table retrieval. In widely used embedding evaluation benchmarks such as MMEB, we observe that they include some non-natural images, like figures and tables. However, their scale is typically small and fails to reflect the performance of embedding models in real-world scenarios. Therefore, we propose TaR-ViR as a complementary evaluation benchmark for existing embedding models.
>
> For Question 1:
>
> When all images are converted into handwritten or low-quality scanned data, we thought the performance would decline.  Before developing TaR-ViR, we attempted to generate handwritten images from text using diffusion models. However, these models often fail to adhere strictly to instructions, leading to loss of textual content and inconsistent drawing styles that undermine table integrity. To ensure data completeness, we instead collected corresponding screenshots from Wikipedia. Establishing a comprehensive, real-world table retrieval dataset remains our long-term objective.
>
> For Question 2:
>
> When trained solely on TaR-ViR, the retriever can be deployed on a corpus composed of tables across different modalities. However, its performance remains below those trained on a single, homogeneous corpus. Real-world applications involve complex, mixed-format corpora, presenting a key challenge. To address this, we argue that competitive retrieval necessitates a diverse training set, enabling models to learn unified representations across different data modalities.
>
> For Question 3:
>
> This is an important question. We thought that both generating text from table images and getting their representations via a multimodal retriever count as one MLLM invocation. For OCR-based retrieval, the process involves two steps: first, using an OCR model to extract text content, then invoking a text retriever to generate representations. Thus, OCR-based methods require twice as many model invocations as multimodal methods. However, in practice, the OCR model and text retriever are often separate, so efficiency cannot be judged solely by invocation count. In our experiments, we used Qwen2.5-VL 7B for OCR to convert images to HTML tables. Results show that using one powerful, small embedding model can achieve competitive performance, and its inference overhead is less than the OCR process. Therefore, we conclude that under similar computational overhead, OCR-based text retrieval is less effective than multimodal retrieval.

---

> > ### Comment · Reviewer_5rq1 · 2025-11-27
> >
> > Your rebuttal effectively addressed most of my concerns. One remaining area where further clarification would be valuable is whether you have analyzed performance differences under varying levels of data diversity or noise. Providing some quantitative evidence here would help demonstrate the method’s robustness in real-world scenarios. Overall, your responses strengthened the paper, and I am raising my score to 6.

---

### Official Review · Reviewer_g7XG · 2025-11-09

**Soundness:** 3
**Presentation:** 2
**Contribution:** 2
**Rating:** 4
**Confidence:** 4

**Summary:**

This paper targets at open-domain table retrieval. Instead of using text as the source of retrieval, this paper proposes to use image as an alternative. This paper introduces a new benchmark TaR-ViR based on an existing text-based table benchmark. The authors have conducted some ablation studies that show how the table image would help in scenarios like RAG setup.

**Strengths:**

- Though treating tables as images has been explored in various existing literature (see weakness 3), the authors introduce this setup in the table retrieval.

- The authors have conducted experiments to show the potential of treating tables as images.

**Weaknesses:**

- I wonder if the dataset is comprehensive enough to cover diverse visual web table types. As stated by the authors, `...adapt a textual table retrieval dataset.`, the authors may ignore the types of tables explored in [11]. I believe this distinguishes the visual table understanding / retrieval from considering tables just from the text perspective.

- [12] has conducted ablations on different resolutions of table images, [1] has proposed together with different table formats in text, different image formats for tables. It would be nice for the authors to conduct similar ablation studies to understand how these factors play in your setup.

- In certain experiments, the size and the type of models used are limited. For instance, in the RAG setup, the authors mostly conducted their experiments on 7-8B sized LLMs with 2B retrievers.

- Related works such as [1] is worth mentioning. In the related work section, it is worth mentioning more efforts from the table community, for instance, the ones working on architecture changes [2], the recent waves on instruction tuning foundational LLMs on tables, including [3, 4, 5, 6, 7, 9], and the line of research on investigating table representations [8, 10].

### References

[1] Naihao Deng, Zhenjie Sun, Ruiqi He, Aman Sikka, Yulong Chen, Lin Ma, Yue Zhang, and Rada Mihalcea. 2024. Tables as Texts or Images: Evaluating the Table Reasoning Ability of LLMs and MLLMs. In Findings of the Association for Computational Linguistics: ACL 2024, pages 407–426, Bangkok, Thailand. Association for Computational Linguistics.

[2] Jingfeng Yang, Aditya Gupta, Shyam Upadhyay, Luheng He, Rahul Goel, and Shachi Paul. 2022. TableFormer: Robust Transformer Modeling for Table-Text Encoding. In Proceedings of the 60th Annual Meeting of the Association for Computational Linguistics (Volume 1: Long Papers), pages 528–537, Dublin, Ireland. Association for Computational Linguistics.

[3] Tianshu Zhang, Xiang Yue, Yifei Li, and Huan Sun. 2024. TableLlama: Towards Open Large Generalist Models for Tables. In Proceedings of the 2024 Conference of the North American Chapter of the Association for Computational Linguistics: Human Language Technologies (Volume 1: Long Papers), pages 6024–6044, Mexico City, Mexico. Association for Computational Linguistics.

[4] Xiaokang Zhang, Sijia Luo, Bohan Zhang, Zeyao Ma, Jing Zhang, Yang Li, Guanlin Li, Zijun Yao, Kangli Xu, Jinchang Zhou, Daniel Zhang-Li, Jifan Yu, Shu Zhao, Juanzi Li, and Jie Tang. 2025. TableLLM: Enabling Tabular Data Manipulation by LLMs in Real Office Usage Scenarios. In Findings of the Association for Computational Linguistics: ACL 2025, pages 10315–10344, Vienna, Austria. Association for Computational Linguistics.

[5] Naihao Deng and Rada Mihalcea. 2025. Rethinking Table Instruction Tuning. In Findings of the Association for Computational Linguistics: ACL 2025, pages 21757–21780, Vienna, Austria. Association for Computational Linguistics.

[6] Li, Peng, et al. "Table-gpt: Table fine-tuned gpt for diverse table tasks." Proceedings of the ACM on Management of Data 2.3 (2024): 1-28.

[7] Zha, Liangyu, et al. "Tablegpt: Towards unifying tables, nature language and commands into one gpt." arXiv preprint arXiv:2307.08674 (2023).

[8] Li, Liyao, et al. "Table as a Modality for Large Language Models." The Thirty-ninth Annual Conference on Neural Information Processing Systems.

[9] Su, Aofeng, et al. "Tablegpt2: A large multimodal model with tabular data integration." arXiv preprint arXiv:2411.02059 (2024).

[10] Long, Lin, et al. "Bridging the Semantic Gap Between Text and Table: A Case Study on NL2SQL." The Thirteenth International Conference on Learning Representations. 2025.

[11] Titiya, Prasham Yatinkumar, et al. "MMTBENCH: A Unified Benchmark for Complex Multimodal Table Reasoning." arXiv preprint arXiv:2505.21771 (2025).

[12] Mingyu Zheng, Xinwei Feng, Qingyi Si, Qiaoqiao She, Zheng Lin, Wenbin Jiang, and Weiping Wang. 2024. Multimodal Table Understanding. In Proceedings of the 62nd Annual Meeting of the Association for Computational Linguistics (Volume 1: Long Papers), pages 9102–9124, Bangkok, Thailand. Association for Computational Linguistics.

**Questions:**

See weaknesses.

---

> ### Author Response · Authors · 2025-11-24
>
> We sincerely appreciate your diligent review and thoughtful feedback. Below is our point-by-point response to the specific comments you provided.
>
> For Weakness 1:
>
> Table style is an important factor to consider when constructing the TaR-ViR. To be precise, we utilized the relevance annotations of NQ-TABLES. During the process of collecting tables from Wikipedia, we observed that tables within the main text are presented in a variety of formats. Additionally, infoboxes and navigation bars are also organized in tabular format, which was also preserved during our collection process. We thought the styles of the table images in TaR-ViR were diverse. During the collection process, we found that most of the table types mentioned in [1] are all included in TaR-ViR. However, due to the large volume of tabular data contained within TaR-ViR, we found it challenging to accurately classify them into their corresponding table categories. Therefore, we have not shown statistical results for different table styles.
>
> For Weakness 2:
>
> This is a very valuable suggestion. Resolution is a very important factor affecting multimodal retrieval. We analyzed the table images saved in TaR-ViR and found significant resolution variations among these table images. Using browser window size as the classification criterion, we categorized all table images into Easy (<1/4 browser window), Middle (1/4–1 browser window), and Hard (>1 browser window), as shown in Figure 2. Furthermore, we have compared the performance of table images at different resolutions in Table 5. We found that as tables grow in both structural complexity and physical dimensions, MLLMs struggle to accurately extract and parse all textual content appearing in tabular images. The image-centric table retrieval paradigm is gradually demonstrating its advantages. In Table 4, we compared the performance of the retrievers under different input formats based on Qwen2-VL. Since we used table screenshots collected from Wikipedia, we have limited editing options for the table styles. We explored different formats of text input, including OCR-based text and HTML-based text.
>
> For Weakness 3:
>
> Let me elaborate further on the principles behind some of our experimental settings. We compared the performance of two retrieval paradigms in the context of table retrieval. For text-based retrievers, we utilized Qwen2.5-VL 7B to extract and organize table content. Then we will use the extracted content as input for text retrievers. The text retrieval paradigm involves the process of extracting table content from images using MLLMs, which has a high inference cost. The MLLM-based multimodal retriever eliminates the need for extracting table content from images. To compare the performance of two different retrieval paradigms at comparable inference costs, we selected some small-sized text retrievers. Comparing the retrieval results, we found that multimodal retrieval has advantages in both efficiency and effectiveness when resolving table retrieval tasks. Furthermore, we would like to verify whether storing tables as images also offers advantages in RAG scenarios. Therefore, we selected a set of LLMs/MLLMs of similar size for comparison.
>
> For Weakness 4:
>
> Research progress in the field of tables relies heavily on the efforts and contributions from the community. Due to space constraints, our previous version primarily covered retrieval-based related work. In subsequent revisions, we have incorporated these works into our related work section as per your suggestion to provide a more comprehensive overview.
>
> [1] Titiya, Prasham Yatinkumar, et al. "MMTBENCH: A Unified Benchmark for Complex Multimodal Table Reasoning." arXiv preprint arXiv:2505.21771 (2025).

---

> > ### Comment · Reviewer_g7XG · 2025-11-28
> > **Response**
> >
> > - W 1, I was talking about [11] instead of [1].
> >
> > - W 2, I was talking about resolution and different table representations (e.g., markdown, html, json, etc). What you posted here is not relevant.
> >
> > - W 3, I still feel 2B sized retrievers may not lead to a solid conclusion here.
> >
> > - W 4, thank you.

---

> ### Author Response · Authors · 2025-11-28
>
> Thank you for your response. Regarding these weaknesses, I have the following additional explanation:
>
> For Weakness 1:
>
> Concerning the paper “MMTBENCH: A Unified Benchmark for Complex Multimodal Table Reasoning,” which you cited as reference [11] in your comments and I cited as reference [1] in my reply, I apologize for some errors in my response and have corrected them.
>
> For Weakness 2:
>
> The paper and previous responses primarily demonstrate that table images of different resolutions have different performances in multimodal table retrieval. Following the approach outlined in Table-LLaVA for analyzing table images, we resize images to three resolution levels as defined by VLM2VEC: high (1344×1344), medium (672×672), and low (128×128). We trained Qwen2-VL 2B using table images at different resolutions, resulting in several different variants. Their corresponding retrieval performance is shown in the table below:
>
> | Backbone    | Resolution | R@10  | N@5   | M@5   |
> |-------------|------------|-------|-------|-------|
> | Qwen2-VL 2B | low        | 88.38 | 56.13 | 49.56 |
> | Qwen2-VL 2B | middle     | 90.36 | 64.59 | 59.01 |
> | Qwen2-VL 2B | high       | 91.93 | 67.73 | 62.39 |
>
> We can observe that as the resolution of table images gradually increases, multimodal retrievers are able to understand their content accurately, thereby enhancing retrieval performance. As image resolution increases, the encoding overhead also increases accordingly. The selection of image resolution for tables requires a balance between effectiveness and efficiency. Additionally, we have analyzed the impact of different formats in Table 4.
>
> For Weakness 3:
>
> We agree with your concerns. Therefore, we supplemented the results of the Qwen2-VL 7B model with the same experimental setup as described in the paper. We trained the Qwen2-VL 7B as a retriever using different inputs (text and multimodal data) to obtain its corresponding text retriever and multimodal retriever. Furthermore, we have also validated its performance using RAG. The results are shown in the following table.
>
> | Retriever                          | Generator                       | n=1   | n=3   | n=5   |
> |------------------------------------|---------------------------------|-------|-------|-------|
> | Qwen2-VL 7B (Text Retriever)       | Mistral-7B                      | 30.94 | 34.93 | 35.83 |
> | Qwen2-VL 7B (Text Retriever)       | Qwen3-8B                        | 44.13 | 56.51 | 58.95 |
> | Qwen2-VL 7B (Text Retriever)       | Qwen2.5-VL 7B (Text Input)      | 32.65 | 38.59 | 39.08 |
> | Qwen2-VL 7B (Text Retriever)       | LLaVA-OneVision                 | 19.95 | 22.71 | 19.44 |
> | Qwen2-VL 7B (Text Retriever)       | Qwen2.5-VL 7B (MultiModal Input)| 33.31 | 40.99 | 42.61 |
> | Qwen2-VL 7B (MultiModal Retriever) | Mistral-7B                      | 31.59 | 35.42 | 37.29 |
> | Qwen2-VL 7B (MultiModal Retriever) | Qwen3-8B                        | 44.62 | 56.35 | 59.52 |
> | Qwen2-VL 7B (MultiModal Retriever) | Qwen2.5-VL 7B (Text Input)      | 33.38 | 39.65 | 41.04 |
> | Qwen2-VL 7B (MultiModal Retriever) | LLaVA-OneVision                 | 21.57 | 23.12 | 20.01 |
> | Qwen2-VL 7B (MultiModal Retriever) | Qwen2.5-VL 7B (MultiModal Input)| 34.36 | 41.43 | 42.77 |
>
> When the retriever is replaced with Qwen2-VL 7B, the difference between multimodal retrieval and text retrieval becomes more pronounced. When the results from the multimodal retriever are used as input for LLMs and MLLMs, the generated answers are better than when text retriever results are used as input. This further demonstrates the advantages of treating table retrieval as multimodal retrieval.
>
>
> Thank you again for your discussion.

---

### Author Response · Authors · 2025-12-03
**Summary of Rebuttal**

Dear Area Chair,

We sincerely thank the reviewers for their thoughtful feedback. We are encouraged to see that the reviewers have acknowledged the soundness and value of TaR-ViR. Initially, this paper received scores of **8, 6, 4, 4, and 4**. We want to thank reviewer U5D7 and reviewer LgPw for their recognition of our work.  Following the initial discussion, reviewer 5rq1 considered that our rebuttal addressed the concerns and **raised the score from 4 to 6**. Reviewer g7XG **raised new questions in response to our reply, and we have provided further explanations**. However, due to the early termination of the discussion, we were unable to continue our discussion.
Nevertheless, we believe our response addresses the concerns of reviewer g7XG. And the remaining reviewers did not participate in the discussion, but we believe that our responses adequately address their concerns. Below, we provide a concise summary of our rebuttal and clarifications to support your final assessment.

**Consensus Among Reviewers**
- The Scale of TaR-ViR: Compared to other datasets, TaR-ViR incorporates numerous tables, reflecting the practical application scenarios of tables with different styles.
- Comprehensive Experiments: We selected a wide range of retrievers for comparison and introduced RAG evaluation to demonstrate the potential value of accomplishing table retrieval through multimodal retrieval.
- Efficiency and Cost-Effectiveness of Data Annotation: We designed an annotation-assisted pipeline that completed the construction of the TaR-ViR with a small number of human annotators.

**Summary of Clarifications**
- Table Style: Since TaR-ViR is collected from Wikipedia webpages, the diversity of table styles is an important consideration. We would like to explain that due to Wikipedia's varied content presentation, table styles are diverse. Consequently, the TaR-ViR dataset exhibits diversity and can represent tables in real-world applications.
- Evaluation of RAG: In our response to the reviewers, we explained the purpose of introducing RAG: to demonstrate the impact of multimodal retrieval on downstream question-answering performance by comparing results from fair text retrievers and multimodal retrievers. Additionally, beyond the experiments with Qwen2-VL 2B presented in the main text, we supplemented our findings with experiments using the Qwen2-VL 7B model.
- Supplement for the ablation experiments: Considering the problems of missing information and resolution during table collection, we added ablation experiments with low resolution and missing titles to show the performance of multimodal retrieval under different settings for table retrieval.

We hope the Area Chair will consider our rebuttal and the corresponding clarifications. We believe these additions adequately address the reviewers’ concerns.

Thank you again for your time and effort!

Best regards,

Authors

---

### Meta-Review · Area_Chair_9rUT · 2026-01-05

**Summary:**

This paper introduces TaR-ViR, a multimodal benchmark that redefines open-domain table retrieval by treating tables as images rather than text sequences. The work argues that text-only approaches fail to capture the rich structural and spatial semantics of real-world tables. TaR-ViR extends the NQ-TABLES dataset by collecting approximately 2 million table screenshots from Wikipedia and aligning them with natural-language queries via a semi-automated annotation pipeline that leverages MLLMs. Comprehensive experiments compare text-based retrievers against multimodal ones, showing that multimodal retrievers achieve competitive or superior performance, particularly in recall and large-scale retrieval efficiency.

The paper received five reviews, and the authors did their best to address the reviewers' comments. The proposed dataset will be useful in the community, although to avoid any bias experiments should be reported on models other than Qwen (e.g., Gemma), since Qwen was used in the creation of the dataset. I would also encourage the authors to cite this paper:

https://arxiv.org/abs/2406.08100

**Reviewer Concerns:**

I summarize the concerns below and how they were addressed:

1.  Multiple reviewers questioned whether TaR-ViR provides sufficient diversity, noting it relies exclusively on Wikipedia sources with "relatively clean, consistently formatted" tables that may not reflect real-world heterogeneity, including scanned documents, enterprise spreadsheets, and non-Wikipedia layouts.

The authors clarified that Wikipedia tables exhibit substantial format diversity due to varied content presentation methods, including infoboxes, navigation bars, and multiple table styles within individual webpages. They acknowledged the Wikipedia-centric limitation while emphasizing that cost and scale considerations necessitated this approach, noting TaR-ViR contains significantly more tables than manually constructed datasets.

2.  Reviewers requested comprehensive ablation studies examining resolution effects, different table representations (mark down, HTML, JSON), and performance degradation when critical metadata (e.g., titles) is unavailable.

The authors substantially expanded their experimental analysis by adding: (1) resolution ablations demonstrating performance across low (128×128), medium (672×672), and high (1344×1344) resolutions; (2) table format comparisons including OCR-based and HTML-based text inputs; (3) missing title experiments showing significant performance degradation when titles are unavailable, with specialized encoders demonstrating better robustness.

3. Experiments primarily utilized small models (2B retrievers, 7-8B LLMs), raising questions about whether conclusions would generalize to larger, more capable models.

The authors supplemented their results with Qwen2-VL 7B experiments under identical experimental settings, demonstrating that multimodal retrieval advantages become more pronounced with larger models. They explained the initial model selection was motivated by computational efficiency comparisons: multimodal retrieval eliminates the need for separate OCR and text retrieval steps required by text-based approaches.

4. The semi-automated annotation pipeline using Qwen2.5-VL-72B achieved 80% precision in human verification, suggesting non-trivial label noise that could introduce training bias, particularly since multimodal retrievers were trained on partially machine-labeled data.

The authors acknowledged the trade-off between dataset scale and annotation precision, emphasizing that TaR-ViR prioritizes real-world application scenarios requiring large-scale data. They implemented a hybrid strategy: comprehensive manual annotation for the test set to ensure accuracy, while leveraging MLLM assistance for the training set.

5. The retrieval-augmented generation (RAG) experiments showed only marginal improvements, with text-only LLMs still achieving superior performance, raising questions about the practical value of the multimodal retrieval paradigm for downstream tasks.

The authors clarified that RAG experiments were designed to demonstrate whether multimodal retrieval advantages persist in downstream applications rather than to optimize QA performance.. They attributed poor MLLM QA performance to current models being primarily trained on natural images, identifying this as a key optimization opportunity rather than a limitation of the multimodal retrieval paradigm.

**Reviewer Scores:**

Initial reviewer  scores were: 8, 6, 4, 4, 4
After the rebuttal, one reviewer (5rq1) raised their score from 4 to 6. However, two reviewers (score 4) either didn't respond or had ongoing concerns about dataset diversity, model sizes, and novelty.

---

### Decision · Program_Chairs · 2026-01-26

Accept (Poster)